# Sn1,3 Regiospecificity of DHA (22:6ω-3) of Plant Origin (DHA-Canola^®^) Facilitates Its Preferential Tissue Incorporation in Rats Compared to sn2 DHA in Algal Oil at Low Dietary Inclusion Levels

**DOI:** 10.3390/nu17081306

**Published:** 2025-04-09

**Authors:** Damien P. Belobrajdic, Julie A. Dallimore, Michael J. Adams, Surinder P. Singh, Mahinda Y. Abeywardena

**Affiliations:** 1CSIRO Health and Biosecurity, Adelaide, SA 5000, Australiajuliedallimore68@gmail.com (J.A.D.);; 2Alliance for Research in Exercise, Nutrition and Activity (ARENA), University of South Australia, Adelaide, SA 5001, Australia; 3Medicine and Public Health, Health Flinders University, Adelaide, SA 5042, Australia; 4CSIRO Agriculture and Food, Canberra, ACT 2601, Australia; surinder.singh@csiro.au

**Keywords:** DHA, DPA, EPA, omega-3 index, Canola, fish oil, Krill oil, ω-3 LCPUFA, ω-3 fatty acids

## Abstract

Background/Objectives: Regiospecificity in triacylglycerols (TAGs) influences absorption/bioavailability of dietary fatty acids. We evaluated whether sn1,3 located DHA (22:6ω3) of a transgenic higher plant (DHA-Canola^®^) preferentially facilitates its tissue incorporation as compared to sn2 positioned DHA (DHASCO^®^ of algal origin). Methods: Sprague Dawley rats were fed diets (12 weeks) containing DHA-Canola or DHA-Control (a blend of DHASCO^®^ and high oleic sunflower seed oil (HOSO)) at 0.3%, 1%, 3%, and 6% (*w*/*w*), or 7% HOSO prior to determination of tissue fatty acids. Results: At 0.3 and 1% *w*/*w* supplementation, plasma, liver and cardiac tissue DHA incorporation was higher in the plant-based oil (DHA-Canola vs. DHA-Control; *p* < 0.05), whilst sn2 enriched algal oil yielded better outcomes at higher doses (at 3% inclusion, plasma values were 7.8 vs. 5.9%, and at 6% supplementation, 10.0 vs. 7.9 in favor of DHA-Control, *p* < 0.05) At lower intakes, sn1,3 regiospecificity (DHA-Canola) increased the omega-3 index, a clinically relevant biomarker, compared to DHA-Control (*p* < 0.05). Similarly, a build-up of 20:5ω3 and 22:5ω3 occurred with DHA-Canola. Consequently, total omega3s were higher in this latter group. Conclusions: At lower intakes, sn1,3 regiospecificity of DHA leads to its preferential tissue incorporation compared to sn2 DHA.

## 1. Introduction

ω-3 long chain (C ≥ 20) polyunsaturated fatty acids (LCPUFA) derived from marine sources (seafood, fish and microalgae) are well recognized as affording cardiovascular protection [1,2], namely preventing congestive heart failure as well as vulnerability to cardiac arrhythmias and sudden cardiac death [3]. The dietary intake of preformed ω-3 LCPUFA, EPA (20:5ω3), and DHA (22:6ω3), has been recognized as important [2] since in vivo conversion in humans of the shorter-chain (C18) precursor fatty acid, namely α-linolenic acid (ALA, α18:3 ω-3) to DHA, is relatively poor [4,5].

Due to the pleiotropic nature of EPA and DHA, global agencies such as the WHO, FAO, and the 2010 Dietary Guidelines of the United States recommend a daily intake of at least 250 mg EPA + DHA to maintain good health [6], whereas for the secondary prevention of coronary artery disease, the American Heart Association guidelines stipulate an intake of 1 g/d of EPA + DHA via daily consumption of fish, or alternatively, by taking fish oil supplements [7]. However, these recommended intake levels are much higher than the current consumption level in the US, which is estimated to be about 110 mg/day [7,8]. Importantly, there are major limitations in the supply, cost, and safety of providing the amount of fish and fish oil products that are required to meet the recommended levels of intake for improved health. For instance, the global production of fish oil is about 1 million tonnes per annum and has remained fairly steady for the last decade. Although commercial sources of ω-3 LCPUFA, such as micro algal-based oils (e.g., Ulkenia sp. SAM2179, a thraustochytrid) and from krill harvested from Antarctic waters, have increased in recent years, continued expansion of harvest has the potential to adversely affect global fish stocks as these tiny crustaceans are near the bottom of the aquatic food chain. Furthermore, it has been realized for some time that the world fish stocks, both wild and farmed, are insufficient to meet the rising global demand for ω-3 LCPUFA [9], prompting scientists to explore alternative means of producing these bioactive fatty acids. In this regard, genetically modified yeast and transgenic land-based plants have also been proposed as potential future sources of ω-3 LCPUFA [6,10,11,12,13,14,15].

We have recently demonstrated the successful synthesis of both EPA and DHA in several crop plants [6,10,12,16,17,18,19]. DHA levels comparable to those found in fish (up to 15% *w*/*w*) have been achieved in the seeds of the model plant Arabidopsis thaliana [20,21] as compared to 12% of DHA, which is generally found in bulk fish oil. These initial observations have recently been extended to include a common seed oil crop, a Canola variety, that produces high levels of DHA. A unique characteristic of the DHA-Canola is that a higher proportion of the bioactive fatty acid DHA is accumulated in the outer (sn-1,3) position(s) of the triacylglycerol (TAG) molecule, rather than the middle (sn-2) position in TAGs for fish and DHA-rich single cell oil (DHASCO; [18,19,20,21,22]. In a preliminary rat feeding study using a pair of regio-isometrically pure TAG containing DHA positioned at either sn2 or sn1(3) showed that DHA incorporation into cardiac tissue was preferential and dose related [23]. It is noteworthy that regiospecificity is an important determinant of the nutritional outcomes of dietary fatty acids [24,25]. For instance, TAG structure influences short-term intestinal absorption/bioavailability and has long-term effects emanating from the absorbed fatty acids. Fatty acids at sn-1,3 positions are hydrolyzed by pancreatic lipases and absorbed as free fatty acids, whilst sn-2 monoacylglycerols are taken up intact and may influence physiological outcomes [24]. Therefore, strategic positioning of bioactive fatty acids such as ω-3 LCPUFA may influence not only oxidative stability, but also absorption, tissue uptake, and downstream metabolic activity. Indeed, our previous work using a chemically synthesized model TAGs has suggested improved DHA uptake when positioned at sn-1,3 [23].

The present study is the first animal trial to evaluate the ω-3 LCPUFA bioavailability of a plant-based DHA oil (DHA-Canola, with a higher proportion of DHA located at sn-1,3) compared with a DHA-rich algal oil (DHASCO) with DHA primarily occupying the sn-2 position of the TAG structure. DHA is preferentially positioned (98%) at sn-1,3 in DHA-Canola oil, whereas >50% of the DHA in DHASCO oil TAGs occupies the sn-2 position in TAGs. A range of doses were tested to examine the relationship between DHA content in the diet and tissue incorporation so that possible differences in DHA bioavailability arising from regiospecificity can be investigated. It is known that ω-3 LCPUFA are preferentially absorbed compared with ω6 PUFA and, therefore, may be effective at low dietary intakes (<0.5% *w*/*w*). Accordingly, we evaluated a dose–response relationship ranging from the lowest dose (0.3%) known to affect tissue incorporation to a high dose (6%) that is bioequivalent to the highest dose that might be consumed by people (~5 g DHA/d) seeking therapeutic benefits. The diets were fed to rats for a period of 12 weeks; DHA fatty acid absorption was measured by quantifying plasma fatty acid composition, and tissue fatty acid incorporation was quantified in red blood cells (RBC), liver, heart, kidney, skeletal muscle, brain, and testes. It was hypothesized that DHA-Canola will have greater ω-3 LCPUFA bioavailability when compared to a commercial marine-based oil.

## 2. Materials and Methods

### 2.1. Animals

The use of animals in the present study was approved by the CSIRO SA Animal Ethics Committee. All experimental procedures, including the care, handling, and maintenance of the animals, were performed according to the NHMRC guidelines for the use and care of animals for experimental purposes. The sample size was determined a priori, using data from a previous study in which rats of the same strain were fed 1.5% fish oil and DHA in the sn1/sn3 positions (unpublished). Differences in liver and plasma DHA levels were used to determine that a sample size of 6–8 animals would provide >80% power of detecting a difference between the two groups (*p* < 0.05).

In total, 72 male Sprague Dawley rats at 4 weeks of age were obtained from the Animal Resource Centre (Perth, WA, Australia) and established in the animal facility. Animals were fed a semisynthetic AIN 93M diet containing 7% oil from high oleic sunflower oil (HOSO, Cargill Australia, Melbourne, VIC, Australia) for a period of 4 weeks. At 8 weeks of age, animals were randomly allocated via body weight and a randomized complete block design to HOSO or four different dose levels of DHA-Canola oil (CSIRO Werribee, VIC, Australia) or DHA-Control (DHASCO, DSM Nutritional products, Kingstree, SC, USA). Animals in the DHA-Canola oil and DHA-Control groups were provided with 0.3%, 1.0%, 3.0%, or 6.0% (*w*/*w*) of the respective oils, with the balance of oil from HOSO. All diets contained a total of 7.0% (*w*/*w*) oil.

Animals were housed in solid-based boxes containing environmental enrichment and conventional bedding in groups of 4 (n = 8 per diet treatment group), with bodyweight measured weekly and feed intake measured daily. The animal boxes were positioned in racks based on computer-based random sequence generation. Food was provided fresh daily to minimize any possible effect of oxidation of the test oils. Uneaten food was removed from the cage and disposed of. Animals consumed a standard AIN 93M diet (wash out) without fish oil supplementation for a period of 4 weeks prior to commencement of the test diets. Test diets were fed for a period of 12 weeks, and there were no adverse unexpected adverse events during this time. The animals were then killed for tissue collection and subsequent analysis.

At the completion of the study, the rats were anesthetized using sodium pentobarbitone (60 mg/kg). Animals then underwent a dual energy X-ray absorptiometry (Lunar Prodigy, GE Lunar, Madison, WI, USA) scan, and Encore software (Version 13.60) was used to determine fat, lean tissue and bone mass. The rats were killed by exsanguination of the abdominal aorta, and the blood was collected in EDTA vacutainers and centrifuged at 1500× *g* for 10 min. Plasma was collected and stored at −80 °C, and the red blood cells were rinsed in isotonic saline before ultra-pure water was added to lyse the cells. Samples were snap frozen in liquid nitrogen prior to storage at −80 °C for fatty acid analysis. Major organs and adipose tissue depots were removed and weighed. Visceral adipose tissue depots included mesenteric, epididymal, and retroperitoneal fat. Total visceral fat mass was reported as the sum of these fat pad weights. Inguinal fat was weighed as a region of subcutaneous fat. All other tissues were snap frozen in liquid nitrogen and stored at −80 °C for fatty acid analysis.

### 2.2. Diets

The experimental diets were formulated according to AIN-93M (Appendix A) and all ingredients were food grade. Casein was purchased from Rogers and Co. (Melbourne, VIC, Australia), and corn flour and sugar were purchased from PFD Foods Australia. Vitamins and minerals were sourced from MP Biomedicals, and the remaining products, except the experimental oils, were purchased from Sigma Aldrich (Melbourne, VIC, Australia). Diets were manufactured in-house, dried overnight at 40 °C, and stored frozen −20 °C.

The fatty acid composition of the test oils are shown in Table 1, and the levels of DHA added to the diets were chosen based on the different human equivalent doses recommended by the major global health authorities. The diets containing 0.3, 1, 3, and 6% of the test oils were equivalent to a 75 kg person consuming 0.3, 0.8, 2.4, and 4.9 g of DHA per day. The equivalent animal doses were calculated based on allometric scaling and body surface area-based calculations considering the human equivalent dose in a normal-weight adult (70 kg) with the use of the following formula: Human equivalent dose (mg/kg) = Animal dose (mg/kg) × [rat Km/human Km) and assuming a daily food intake of 20 g/d [26].

### 2.3. Biochemical Analyses

Serum total cholesterol, triacylglycerols, and glucose were determined using relevant reagents and protocols provided by Beckman Coulter and quantified using the Beckman AU480 chemistry analyser (Melbourne, Victoria, Australia).

The fatty acid profile of the phospholipid fraction was determined for heart, liver brain, testes, skeletal muscle, and kidney, whereas the total fatty acids were measured in plasma and red blood cells. The phospholipid fatty acid fractions were extracted from samples via a solid phase extraction followed by gas chromatography analysis of fatty acid methyl esters (FAME), as described previously [27]. The FAME were quantified by gas chromatography. A 1 μL sample was injected onto a bonded-phase vitreous-silica DB-23 column (60 m × 0.25 mm × 0.15 µm; SGE) in an Agilent 6890 N gas chromatograph equipped with a split (50:1) injector. The oven temperature program was initially set at 130 °C (hold for 1 min), increased to 170 °C (6.5 °C/min), increased to 215 °C (2.75 °C/min, hold for 12 min), and increased to 230 °C (40 °C/min, hold for 3 min), and the flame ionization detector was set at 280 °C. Peak identification was based on a comparison of retention times with Supelco 37-Component FAME Mix 47 885-U (Sigma-Aldrich, Melbourne, VIC, Australia). Individual fatty acids were calculated as a percentage of the total fatty acids. The omega-3 index was calculated by adding the red blood cell content of EPA and DHA and expressing this value as a percentage of total red blood cell fatty acids [28].

### 2.4. Statistical Analyses

The data are presented as the arithmetic means and SEM for each treatment group. Body composition and fatty acid data were analyzed as a randomized complete block design with 2 × 4 factorial treatment structure using a 2-way ANOVA. Significant interactions between test oil type and test oil amount were analyzed using pair-wise comparisons of simple main effects and applying a Bonferroni adjustment for multiple comparisons. In the absence of an interaction, difference between treatments was assessed by a Tukey’s post-hoc test. Fatty acid data was also compared between the test oils at each level of dietary inclusion using a 2-tailed *t*-test. The effects of diet on body weight gain were determined by repeated measures analysis of variance, with differences between treatments analyzed by a Tukey post hoc test. Nonparametric data were log transformed prior to analysis. It was determined a priori that no data collected were excluded from analysis, and data analysis was conducted blinded. These analyses were performed using SPSS version 23.0 (SPSS Inc., Chicago, IL, USA). A value of *p* < 0.05 was taken as the criterion of significance.

## 3. Results

### 3.1. Animal Growth and Feed Intake

Rats in all dietary treatment groups showed appreciable weight gain reflecting normal animal growth, with similar final body weights for all groups (Table 2). The body weight gain and final body weights of DHA-Canola fed animals were similar to animals fed DHA-Control diets. However, body weight gain was higher for the 6% DHA Control group compared to the 3% DHA-Control group (Table 2), but this was not observed for the other diets containing lower levels of DHA-Control or for the DHA-Canola groups (Table 2).

Animals in all dietary treatment groups consumed sufficient levels of feed required for normal growth, and the amount of feed consumed by DHA-Canola (20.7 ± 0.2 g/d) and DHA-Control (20.3 ± 0.4 g/d) fed animals was similar (*p* = 0.463). However, there were some small differences in the amount of feed consumed between some of the dietary treatment groups. Over the 12 weeks of this study, the 6% DHA-Control fed rats consumed more feed (21.5 ± 0.1 g/d) in comparison to 0.3% (20.1 ± 0.1 g/d), 1% (20.0 ± 0.1 g/d) and 3% DHA Control (19.6 ± 0.2 g/d) and 6% DHA-Canola (20.1 ± 0.4 g/d) fed rats. The 3% DHA Control (19.6 ± 0.2 g/d) fed rats also consumed less feed than 1% DHA-Canola (21.1 ± 0.2 g/d) fed rats.

### 3.2. Body Composition

The fat mass of DHA-Canola fed rats (16.5 ± 0.6%) was lower than DHA-Control fed rats (18.9 ± 0.6%, *p* < 0.01). The 1%, 3%, and 6% DHA-Canola fed animals had less fat mass compared to those fed the 6% DHA-Control (Table 2). DHA-Canola fed animals had lower total visceral fat mass (DHA-Control; 4.1 ± 0.1%, DHA-Canola; 3.7 ± 0.1%, *p* = 0.037) in comparison to DHA-Control fed animals (Table 2). Total visceral fat mass tended to be lower for the 6% DHA-Canola compared to 6% DHA-Control, but this was not significant (*p* = 0.088).

The lean mass was greater in the DHA-Canola fed rats (78.6 ± 0.6%) compared to the DHA-Control fed groups (76.2 ± 0.6%, *p* = 0.037). The lean mass of the 1%, 3%, and 6% DHA-Canola and the 3% DHA-Control fed rats was higher than the 6% DHA-Control fed rats (Table 2).

Gross morphological assessment of all major organs, including heart, liver, kidney, brain, spleen, and thymus, was normal (Appendix A). The type and amount of test oil in the diet had no effect on bone mass (Table 2) or major organ weight (Appendix A).

### 3.3. Serum Cholesterol and Lipids

In unfasted animals, serum cholesterol concentration was not affected by the type of oil. Independent of oil type, serum cholesterol was lower at the two highest levels of DHA in the diets; 0.3%, 2.7 ± 0.1 mM; 1%, 2.7 ± 0.1 mM; 3%, 2.3 ± 0.1 mM; and 6%, 2.2 ± 0.1 mM (*p* < 0.0001). Animals fed diets containing 0.3% DHA-Control had higher serum cholesterol compared to animals fed 6% DHA-Canola and DHA-Control at 3% or 6%. Serum cholesterol was also higher for the 1% DHA-Control compared to the 6% DHA-Canola and 3% DHA-Control fed animals (Figure 1A).

Serum triacylglycerol concentration was not affected by the type of oil. However, animals fed diets containing 3% and 6% DHA-Canola had lower serum triacylglycerols compared to animals fed 0.3% or 1% DHA-Canola, 1% DHA-Control, or HOSO diets. The 3% and 6% DHA-Control fed animals only had lower serum triacylglycerols compared to the 1% DHA-Control group (Figure 1B).

### 3.4. Plasma Fatty Acids

Plasma DHA levels were affected by the amount and type of oil in the diet. When diets contained lower levels of the test oils (0.3 and 1%), DHA-Canola fed animals had higher plasma DHA levels than DHA-Control fed animals (Figure 2A). At the 3% inclusion level in the diet, plasma DHA levels were similar for DHA-Canola and DHA-Control, but at 6% inclusion, DHA-Canola fed animals had lower plasma DHA levels compared to animals fed DHA-Control (Figure 2).

At all levels of dietary inclusion, DHA-Canola fed rats had higher plasma DPA and EPA levels than rats fed equivalent levels of DHA-Control (Table 3). The most notable effect was the considerable rise in plasma EPA following supplementation with DHA-Canola. Furthermore, unlike that observed for DHA—where a differential pattern of uptake was evident at the ‘low’ and ‘high’ ends of oil supplementation—the rise in EPA and DPA was dose-related and reflected higher dietary intake. Subsequently, plasma ω-3 LCPUFA levels were higher for DHA-Canola fed rats at all levels of inclusion compared to DHA-Control fed rats (Figure 2B).

### 3.5. Red Blood Cell Fatty Acids

Red blood cell DHA levels were affected by the amount and type of oil in the diet. When diets contained the lowest level of the test oils (0.3%), DHA-Canola fed animals had a higher red blood cell DHA level compared with DHA-Control fed animals. Similar levels of DHA were achieved when the diet contained 1% DHA-Canola or DHA-Control, but at 3 and 6% inclusion in the diet, DHA-Canola fed animals had lower red blood cell DHA levels compared to animals fed DHA-Control (Table 3).

As was the case for plasma, the red blood cell EPA and DPA levels in the DHA-Canola fed rats were higher than DHA-Control, and they continued to increase as the dietary supplementation level increased (Table 3). This observation highlights a key difference in the uptake, further metabolism, and tissue incorporation of ω-3 precursor fatty acids between the test oils.

The omega-3 index improved incrementally as the quantity of DHA-Canola and DHA-Control increased in the diets from 0.3 to 6% (Figure 3). At the 0.3 and 1% levels of inclusion, DHA-Canola showed a higher ω-3 index compared to DHA-Control (Figure 3).

The increase in red blood cell ω-3 LCPUFA with increasing dose of both test oils occurred in conjunction with a reduction in total ω-6 PUFA, which was primarily at the expense of arachidonic acid (20:4 ω-6) (Table 3). Subsequently, the ω-3/ω-6 increased for both test oils as their level of dietary inclusion increased.

### 3.6. Heart Fatty Acids

Heart DHA levels were higher for DHA-Canola compared with DHA-Control when diets contained the lowest level of the test oils (0.3%) (Table 3). Heart DHA levels were similar when the diet contained 1% DHA-Canola or DHA-Control, but at 3 and 6% inclusion in the diet, DHA-Canola fed animals had lower heart DHA levels compared to animals fed DHA-Control (Table 3).

At all levels of dietary inclusion, DHA-Canola fed rats had higher heart EPA and DPA levels than rats fed equivalent levels of DHA-Control (Table 3).

Heart ω-3 LCPUFA (≥20 C) levels for DHA-Canola fed rats were higher at 0.3% and 1% and lower at 6% compared with DHA-Control fed rats (Table 3).

The increase in heart ω-3 LCPUFA with increasing dose of both test oils occurred in conjunction with a reduction in total ω-6 PUFA, which was primarily due to arachidonic acid (20:4 ω-6) (Table 3). Subsequently, the ω-3/ω-6 increased for both test oils as their level of dietary inclusion increased (Table 3).

### 3.7. Liver Fatty Acids

When diets contained lower levels of the test oils (0.3 and 1%), DHA-Canola fed animals had higher liver DHA levels compared with DHA-Control fed animals (Table 3). However, at 3 and 6% inclusion in the diet, DHA-Canola fed animals had lower liver DHA levels compared to animals fed DHA-Control (Table 3).

At all levels of dietary inclusion, DHA-Canola fed rats had higher liver EPA levels than rats fed equivalent levels of DHA-Control, whereas DPA levels were only higher for DHA-Canola when the level of test oil inclusion in the diet was 1% or greater (Table 3).

Liver ω-3 LCPUFA levels were also higher for DHA-Canola fed rats when included at 0.3, 1, and 3% in the diet (but not 6%) compared with DHA-Control fed rats (Table 3).

The increase in liver ω-3 LCPUFA with increasing dose of both test oils occurred in conjunction with a reduction in total ω-6 PUFA, which was primarily due to arachidonic acid (20:4 ω-6) (Table 3). Subsequently, the ω-3/ω-6 increased for both test oils as their level of dietary inclusion increased (Table 3).

### 3.8. Kidney Fatty Acids

Compared to heart and liver tissues, the kidney appears to be inherently low in DHA. Nevertheless, DHA levels increased as the dietary intake of DHA increased. These increases in DHA content were similar for DHA-Canola and DHA-Control, except at the highest dietary supplementation (6%) level, at which DHA levels were lower in the kidney for DHA-Canola fed rats compared to those fed DHA-Control (Table 3).

Kidney EPA levels were higher for DHA-Canola fed rats when included at 0.3, 1, and 3% in the diet (but not 6%) compared with DHA-Control fed rats (Table 3). Although the kidney DPA levels were very low at all levels of dietary inclusion, DHA-Canola fed animals had higher kidney DPA compared with DHA-Control fed animals (Table 3). These increases in ω-3 content of kidney tissue occurred by the displacement of ω6 PUFA content of kidney tissue. This was primarily due to a reduction in arachidonic acid (20:4 ω-6) and resulted in a higher ω-3/ω-6 in animals fed the highest levels of the test oils (Table 3).

### 3.9. Brain Fatty Acids

The EPA, DPA, and DHA content of the brain was unaffected by the amount or type of DHA oil in the experimental diets (Appendix A).

### 3.10. Muscle Fatty Acids

Increasing the level of DHA in the diets increased the muscle DHA level dose-dependently. This increase in muscle DHA level was similar for DHA-Canola and DHA-Control, except for DHA-Canola, which showed DHA incorporation to reach a plateau at the 3% level (Table 3). DHA-Canola fed rats had higher muscle EPA levels than rats fed DHA-Control when the test oils were included at 0.3, 3, and 6% of the diet. The DPA levels in muscle reached a maximal level of inclusion of 1.5% when the diet contained 1% DHA-Canola, whereas the inclusion of DPA only reached 0.8% following the highest dose of DHA-Control (Table 3).

### 3.11. Testes Fatty Acids

Compared to most other tissues, the testes contained rather low amounts of EPA and DHA but was high in DPA of the ω-6 type representing ~20% of the total phospholipid fatty acids (Table 3).

Testes DHA levels were affected by the amount and type of oil in the diet. When diets contained lower levels of the test oils (0.3 and 1%), DHA-Canola fed animals had higher DHA levels compared with DHA-Control fed animals (Table 3). However, at 3 and 6% inclusion in the diet, both groups showed similar levels (Table 3).

## 4. Discussion

The findings of this study clearly demonstrated that plant-based DHA-Canola is bioavailable and leads to similar outcomes as a commercially available marine algal oil high in DHA. DHA-Canola addition to the diet increased plasma ω-3 LCPUFA EPA, DPA, and DHA levels, resulting in the uptake and incorporation of these fatty acids into red blood cells, liver, heart, skeletal muscle, kidney, and testes. Furthermore, the omega-3 index, an emerging biomarker of coronary heart disease risk favorably improved with increasing intake of the DHA containing test oils. For all animals consuming DHA-rich diets, the omega-3 index was above 4%, which was suggested as the minimum level of ω-3 PUFA incorporation that offers cardiovascular protection and reduced risk of death from coronary heart disease [28]. At the higher levels of intake (3% and 6% *w*/*w*), the omega-3 index reached over 9 and ~13% for both test oil preparations.

Based on previous research [23], we hypothesized that positional specificity of DHA within the TAG molecule would influence its uptake and tissue incorporation.More specifically, it was proposed that DHA absorption and tissue incorporation would be greater when DHA occupies the outer (sn-1,3) position(s) of the TAG molecule, as in the case for DHA-Canola (sn-1,3) compared with DHA-Control, a marine algal oil with DHA primarily (53%) occupying the sn-2 position in TAG molecule. In support of our hypothesis, we showed that plasma DHA levels were higher for DHA-Canola compared with DHA-Control at the lowest levels of dietary inclusion (0.3 and 1%), but at the highest level of inclusion (6%), plasma DHA levels were higher for DHA-Control compared with DHA-Canola. A similar observation was made for red blood cell, heart, liver, and testes DHA levels, whereby DHA-Canola showed higher DHA incorporation than DHA-Control at lower supplementation levels (0.3 and/or 1%). Conversely, at the higher intake levels of 3% and 6%, for most tissues studied, greater incorporation of DHA was evident for the marine-based oil compared to DHA-Canola. These findings agree with the established view on fat absorption, which stipulates those fatty acids located on the sn-1,3 position of the TAGs are hydrolyzed by lipases in the gut lumen, transported as phospholipids of chylomicrons, and metabolized independently. Conversely, the sn-2 monoglycerides are absorbed intact and re-assembled to form triacylglycerols [24,25]. It follows that sn-2-monoglycerides (MAGs) from dietary TAGs may provide the backbone for phospholipid synthesis, especially during higher levels of fat absorption (ω-3). The data of the present study can be explained along these metabolic processes in that, at lower dose levels, the fatty acids located at sn-1,3 positions (DHA in DHA-Canola) appear to have been taken up preferentially and incorporated into tissues and/or metabolized. Conversely, at higher intake levels, there would have been an active uptake of sn-2-MAGs (as has been claimed previously), thus explaining the dose-related reversal of DHA incorporation. In addition, brain DHA content was similar for all dietary treatment groups, and position of DHA in TAG did not influence incorporation of DHA into the brain tissue of adult rats.

A higher accumulation of plasma, red blood cell, and liver EPA (20:5 ω-3) for the DHA-Canola group was another notable difference compared to the DHA-Control. The high (~20%) presence of the precursor fatty acid, α-linolenic acid (ALA; α18:3ω-3), in the plant-based oil accounts for the dose-related rise in EPA in the DHA-Canola fed rats compared to the DHA-Control where no ALA was present in the test oil. Therefore, the increased accumulation of plasma and tissue EPA and DHA in the DHA-Canola group would have originated via multiple routes: (1) forward conversion of the precursor fatty acid ALA (18:3 ω-3) to EPA; (2) via further desaturation and elongation of EPA to form DHA; and (3) by direct incorporation of pre-formed DHA that was present in the diet (oil source). In addition, retroconversion of DHA to EPA (via DPA) is also possible, and this latter pathway may be particularly relevant for DHA-Control rats. For example, the kidney EPA levels of 6% DHA-Control rats were 17% higher than HOSO rats, which can only be explained by activation of the fatty acid retro-conversion pathway. It is noteworthy that, compared to the DHA-Canola oil that contained ~20% ALA, the DHA-Control oil (DHASCO/HOSO blend) contained a negligible level of this precursor fatty acid, and furthermore, no pre-formed EPA.

A greater presence of 22:5ω-3 (DPA) was also observed in RBC and most tissues of the DHA-Canola fed rats compared to DHA-Control, especially as the intake levels were increased. It is of interest to note that, of all tissues analyzed, red blood cell showed the greatest enrichment of DPA. In humans, higher circulating and red blood cell DPA levels are positively correlated with lower cardiovascular risk, including lower triacylglycerols, cholesterol, and inflammation [29]. Whilst we did not analyze the fatty acid composition of platelets in the present study, it is of interest to note that platelet DPA levels have been shown to bear a strong negative correlation with mean platelet volume, a risk factor for acute myocardial infarction [30,31,32,33].

Other metabolic effects of DPA have recently been discovered [34,35,36]. A recent study which investigated the postprandial metabolism of DPA and EPA in humans concluded that these two fatty acids undergo different metabolic fates. More interestingly, the presence of DPA (2 g) in a meal containing olive oil (18 g) nearly prevented the incorporation of fatty acids into chylomicrons (the lipoprotein particles that transport lipids from the intestine to other parts of the body), an effect not observed with EPA [30]. One of the potential explanations proposed by the investigators for this action was based on DPA acting as an inhibitor of pancreatic lipase, which is important in the digestion of dietary fat. Inhibitory action on this enzyme can result in impaired digestion of ingested fat, leading to lower absorption and increased excretion. In this regard, it is noteworthy that DHA-Canola was associated with reduced adiposity (lower fat mass) in the present study, which is a highly favorable specific health outcome given that lean (muscle) mass was not different between treatment groups. Taken collectively, these emerging findings implicate diverse health attributes of DPA which warrant further investigation.

The influence of DHA Canola on increasing DHA levels seems to be markedly greater than other high ω-3 transgenic plant-derived oils. A genetically modified soybean preparation containing the shorter (C18) ω-3 PUFA SDA (stearidonic acid 18:4 ω3; 16–28%) increased plasma and red blood cell EPA levels in people, thus increasing their omega-3 index [37,38]. However, the amount of EPA accumulated was relatively small, and DHA levels in plasma and tissue along with blood lipids were unaffected. In support of this, we have shown that there is no endogenous conversion of SDA to DHA in rats, and that any changes in DPA are likely to be of limited cardiac benefit [39]. In this latter study, dietary SDA (as Echium Oil) led to a preferential increase in tissue DPA; however, there was no change in DHA. Therefore, it is clear that novel plant oils (e.g., GM-Soy, Echium) containing C18 ω-3 fatty acids such as SDA (18:4 ω-3) are not able to increase endogenous DHA status since its elongation/desaturation terminates at DPA. Alternatively, a transgenic Camelina satvia seed oil that contains 20% EPA increased hepatic EPA and DHA levels when fed to mice [40]. However, the increase in hepatic DHA levels were up to 50% less than those reported for DHA-Canola in the current study. Additionally, tissue uptake and conversion of EPA into DHA may be limited as muscle DHA levels were not affected by EPA-rich Camelina oil [40], whereas the current study showed that increased muscle DHA levels were directly related to the amount of DHA-Canola in the diet. Further studies also show that dietary EPA only had modest effects on DHA accumulation in mice [41], and when people consumed 1.8 g EPA/d, there was little change in plasma DHA concentration [42].

In the current study, DHA-Canola and DHA-Control showed reductions in serum total triacylglycerols and cholesterol as the dose increased, which was particularly evident at the highest dose. These reductions were less noticeable in hypercholesterolemic rats when a high-oleic sunflower seed oil rich diet was supplemented with 6.6% (*w*/*w*) fish oil, whereas EPA-rich Camelina satvia oil was reported to have no effect on total cholesterol or triacylglycerols in mice [40].

A highly novel and important finding from this study is that DHA-Canola fed animals had significantly less fat mass compared with DHA-Control fed animals. Specifically, the DHA-Canola fed animals had significantly lower visceral fat pad weights, lower total fat mass, and higher lean mass compared to rats fed the DHA-Control diets. As there were no differences in feed intake or feed conversion efficiency between diets containing the different DHA oil sources, this suggests that the oils had contrasting effects on fat metabolism, with DHA-Canola preventing lipogenesis and/or stimulating fatty acid catabolism through beta-oxidation. Conversely, increasing the level of marine-based oil in the diet (DHA-Control) had no effect on adiposity, which is a finding consistent with previous rodent studies when a similar low fat background diet was used [43,44,45], or in the context of diet containing high oleic sunflower seed oil [46]. A reduction in white adipose tissue mass only resulted when the diet was high in fat (20–30% total fat by weight) [47,48,49,50,51,52,53,54]. These studies suggest that the anti-adipogenic effect of LC ω3 PUFA may involve a metabolic switch in adipocytes that includes enhancement of beta-oxidation and upregulation of mitochondrial biogenesis [47], leading to reduced fat cell hypertrophy and hyperplasia [52]. Liver, heart, and skeletal muscle tissue also show an increased expression of mitochondrial uncoupling proteins, which suggests increased fatty acid oxidation in these tissues [51]. However, it remains unclear why DHA-Canola reduced adiposity in a low-fat background diet but DHA-Control did not; this may be due to the regiospecific location of DHA in the sn-2 position of the triacylglycerol molecule, which is the site of preferential absorption [55,56]. Furthermore, emerging evidence suggests DPA as a potential antiobesity agent mediated via inhibition of pancreatic lipase [30]. Considering that one of the key differences between the two test oils is elevation of endogenous DPA level following DHA-Canola, it is worth speculating a unique biological action around fat metabolism for DPA, not shared by the other two ω-3 LCPUFAs. Further studies are warranted to test this possibility.

Our results also showed that, at the highest level of oil inclusion, the DHA-Control diet promoted body weight gain, a finding that is consistent with a study in mice that used EPA-rich Camelina oil or fish oil [40]. Importantly, these effects of DHA dose on body weight gain were not observed for DHA-Canola fed animals. Furthermore, it is unlikely that DHA-Canola contains any anti-nutritive factors such as glucosinolates, as the levels of glucosinolates in the DHA-Canola seed are low and unlikely to be present in the oil as they are highly soluble and removed during oil manufacture.

A strength of using the rat model (Sprague Dawley strain) is that it provides good sensitivity in detecting diet-induced changes in fatty acid bioavailability and tissue uptake. Furthermore, changes in plasma and red blood cells are consistently shown to represent changes seen in clinical trials in humans. Although there are some differences in the digestive tract of human and rats, previous studies have shown that fatty acid digestion/metabolism and tissue incorporation reported in rats can be reliably reproduced in clinical trials. Another strength of this study is that animals were fed the experimental diets for 12 weeks, which is the minimum amount of time required to accurately measure fatty acid incorporation into RBC and enabled us to calculate the omega-3 index, an emerging biomarker of cardiovascular disease. It also provides a sufficient period to demonstrate that the transgenic plant-based Canola oil does not have any adverse effects on general health parameters (growth, gross organ pathology).

## 5. Conclusions

This 12-week feeding trial showed that rats consuming diets containing DHA-Canola had a weight gain, diet intake, and major organ morphology that was reflective of normal, healthy animals of this age. Compared to the DHA-Control diets, DHA-Canola fed rats had lower visceral fat pad weight, lower total fat mass, and higher lean mass, and further studies are warranted to investigate whether these changes are due to the elevated levels of endogenous DPA in these animals. The DHA-Canola fed animals also showed greater absorption and tissue incorporation (heart, liver, testes) of DHA (and all LC ω3 PUFA) at the lowest levels of inclusion (0.3 and 1%) compared with DHA-Control. EPA and DPA levels in plasma, red blood cell, liver, and heart were also higher in DHA-Canola compared with DHA-Control at equivalent dose levels. Although similar increases in the ω3-index were observed as the quantity of both test oils increased in the diets, at the lowest levels of inclusion (0.3 and 1%), the omega-3 index was higher for DHA-Canola compared to DHA-Control. This study suggests that DHA-Canola offers additional benefits to enhancing DHA levels in tissues over a marine-sourced oil when included at low levels in the rat diet, and may provide additional metabolic benefits (lower fat mass) at higher levels of intake. Thus, clinical trials that use equivalent levels of DHA-Canola are warranted to confirm dose and efficacy in people.

## Figures and Tables

**Figure 1 nutrients-17-01306-f001:**
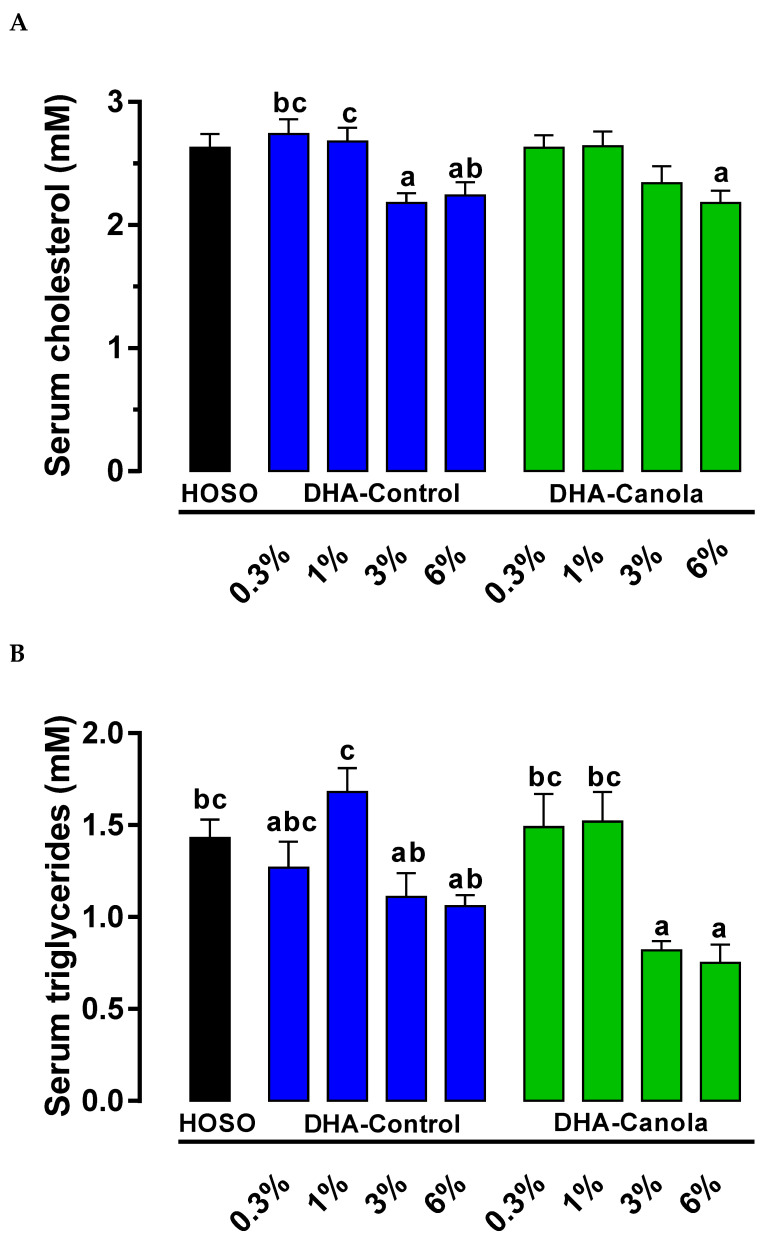
The effect of the Control diet (HOSO) and diets containing four levels of DHA-Control and DHA-Canola on serum (**A**) total cholesterol and (**B**) triacylglycerols in non-fasted male Sprague Dawley rats. Values are expressed as mean ± SEM for n = 7–8 animals per group. Significant differences between dietary groups are indicated by different superscript letters (ANOVA, Tukey’s at *p* < 0.05). DHA, docosahexaenoic acid; HOSO, high oleic sunflower seed oil.

**Figure 2 nutrients-17-01306-f002:**
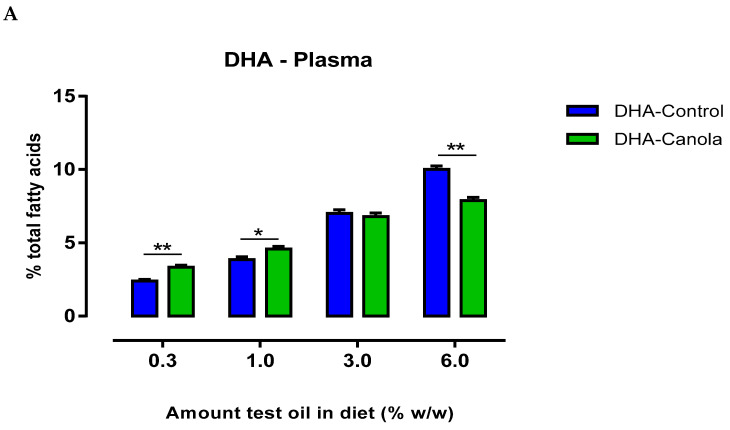
The effect of diets containing four levels of DHA-Control or DHA-Canola on plasma (**A**) DHA level and (**B**) LC ω3 PUFA (≥20C) in male Sprague Dawley rats. Values are expressed as mean ± SEM for n = 7–8 animals per group. At each DHA supplementation level, significant differences between DHA-Control and DHA-Canola are shown as * *p* < 0.05, ** *p* < 0.01 (two-tailed, unpaired student *t*-test). DHA, docosahexaenoic acid.

**Figure 3 nutrients-17-01306-f003:**
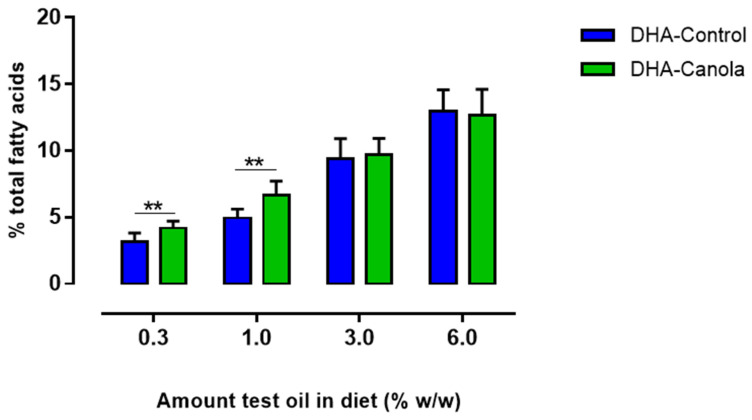
The effect of diets containing four levels of DHA-Control or DHA-Canola on ω-3 index (sum of EPA and DHA). At each DHA supplementation level significant differences between DHA-Control and DHA-Canola are shown as ** *p* < 0.01 (two-tailed, unpaired student *t*-test). Mean values (n = 7–8). DHA, docosahexaenoic acid.

**Table 1 nutrients-17-01306-t001:** Fatty acid composition of test oils (% fatty acids) ^1,2,3^.

Fatty Acid	HOSO	DHA-Control	DHA-Canola
12:0	n.d	0.3 ± 0.0	n.d
14:0	n.d	1.7 ± 0.1	n.d
16:0	4.3 ± 0.2	5.2 ± 0.0	5.0 ± 0.0
16:1 n9	n.d	0.6 ± 0.0	0.2 ± 0.0
18:0	3.6 ± 0.0	2.8 ± 0.0	2.5 ± 0.0
18:1 ω9c	80.6 ± 0.2	72.0 ± 0.2	45.9 ± 0.4
18:2 ω6c	8.7 ± 0.0	7.3 ± 0.0	9.4 ± 0.0
18:3 ω6	n.d	n.d	0.7 ± 0.0
18:3 ω3	n.d	n.d	20.7 ± 0.2
20:0	0.7 ± 0.0	0.3 ± 0.0	1.3 ± 0.0
20:1	0.3 ± 0.0	0.3 ± 0.0	1.5 ± 0.0
20:4 ω6	n.d	n.d	0.7 ± 0.0
22:0	1.2 ± 0.0	0.9 ± 0.0	0.5 ± 0.0
22:1 ω9	n.d	n.d	0.9 ± 0.0
20:5 ω3	n.d	n.d	0.3 ± 0.0
24:0	0.5 ± 0.0	0.3 ± 0.0	0.4 ± 0.0
24:1	n.d	n.d	0.4 ± 0.0
22:5 ω3	n.d	0.1 ± 0.0	0.7 ± 0.0
22:6 ω3	n.d	8.4 ± 0.1	8.7 ± 0.2

^1^ DHA-Control oil is a blend of Docosahexaenoic acid single cell oil (DHASCO) and high oleic sunflower seed oil (HOSO). ^2^ n.d., not detectable. ^3^ Values shown are mean ± SEM of n = 3.

**Table 2 nutrients-17-01306-t002:** Body weight and body composition ^1,2^.

	Final Body Weight (g)	Body Weight Gain (g/wk)	Visceral Fat Pad Weight (%) ^3^	Fat Mass (%)	Lean Mass (%)	Bone Mass (%)
HOSO	496 ± 20.8	19 ± 1.0	4.4 ± 0.3	19.1 ± 1.2	75.9 ± 1.2	12.9 ± 0.6
DHA-Control						
0.3%	504 ± 10.8	19.4 ± 0.6	4.0 ± 0.3	18.6 ± 1.2	75.9 ± 1.1	13.2 ± 0.3
1.0%	503 ± 13.8	19.4 ± 0.5	4.0 ± 0.2	17.6 ± 1.3	77.4 ± 1.4	13.2 ± 0.4
3.0%	493 ± 12.1	18.4 ± 0.8 ^a^	3.7 ± 0.2	17.2 ± 0.7	78.6 ± 0.7 ^d^	12.9 ± 0.4
6.0%	537 ± 13.3	21.8 ± 0.8 ^a^	4.5 ± 0.2	22.3 ± 1.2 ^abc^	72.8 ± 1.1 ^abcd^	14.4 ± 0.5
DHA-Canola						
0.3%	496 ± 7.9	18.7 ± 0.4	4.1 ± 0.3	17.9 ± 1.4	77.3 ± 1.4	12.9 ± 0.3
1.0%	511 ± 18.4	19.9 ± 1.2	3.6 ± 0.3	16.4 ± 1.2 ^a^	78.5 ± 1.1 ^a^	13.4 ± 0.4
3.0%	506 ± 10.7	19.7 ± 0.8	3.7 ± 0.2	16.1 ± 1.1 ^b^	79.7 ± 1.5 ^b^	13.5 ± 0.3
6.0%	499 ± 15.9	18.9 ± 0.9	3.5 ± 0.1	15.5 ± 0.8 ^c^	79.0 ± 0.8 ^c^	13.0 ± 0.4
Oil type	n.s.	n.s.	0.04	0.004	0.005	n.s.
Oil amount	n.s.	n.s.	n.s.	n.s.	0.037	n.s.
Type × amount	n.s.	0.024	n.s.	0.030	n.s.	n.s.

^1^ Values are expressed as mean ± SEM for n = 8 animals per group. ^2^ Significant differences between dietary groups within a column are indicated by common superscript letters (ANOVA, Tukey’s at *p* < 0.05). n.s., not significant; HOSO, high oleic sunflower seed oil. ^3^ Visceral fat mass is the sum of epididymal, mesenteric and peri-renal fat depots.

**Table 3 nutrients-17-01306-t003:** The effect of the Control diet (HOSO) and diets containing DHA-Control and DHA-Canola at four different levels on plasma and tissue fatty acids (only ≥C20 PUFA are shown (% total fatty acids) in rats ^1^.

		0.3%		1.0%		3.0%		6.0%	
	HOSO	Control	Canola	Control	Canola	Control	Canola	Control	Canola
Plasma									
20:4ω6	22.7	22.5	19.5	17.5	17.8 **	12.0	12.6	7.3	9.4 ***
20:5ω3	0.03	0.02	0.4 ***	0.2	1.9 ***	2.3	6.5 ***	5.8	10.5 ***
22:5ω3	0.03	0.1	0.2 *	0.1	0.5 ***	0.4	1.1 ***	0.8	1.5 ***
22:6ω3	1.0	2.4	3.3 **	3.9	4.6 *	7.0	6.8	10.0	7.9 ***
Σ ω3 PUFA	1.0	2.5	4.1 ***	4.3	7.6 ***	10.0	15.7 ***	16.7	22.9 ***
Red blood cell									
20:4ω6	28.8	27.0	26.4	24.5	22.0	19.8	16.8 **	13.7	13.2
20:5ω3	n.d.	0.01	0.2 **	0.2	1.3 ***	1.5	3.8 ***	3.9	6.3 ***
22:5ω3	0.3	0.4	1.1 **	0.9	2.0 ***	1.6	2.8 ***	1.8	3.4 ***
22:6ω3	1.3	3.1	4.0 **	4.8	5.4	7.8	5.9 **	9.1	6.4 **
Σ ω3 PUFA	1.6	3.5	5.3 ***	5.8	8.7 ***	10.9	12.8 *	14.8	16.6
Heart									
20:4ω6	30.0	26.3	23.9 **	20.9	18.1 **	13.7	13.7	9.6	10.2
20:5ω3	n.d.	n.d.	0.06 ***	0.1	0.3 ***	0.4	0.9 ***	1.1	1.5 ***
22:5ω3	0.1	0.4	0.9 ***	0.6	1.6 ***	1.0	2.3 ***	1.1	2.6 **
22:6ω3	4.4	11.0	13.9 ***	17.5	18.9	23.1	20.6 **	27.5	23.7 **
Σ ω3 PUFA	4.9	11.4	14.9 **	18.1	20.9 **	24.4	24.0	29.7	28.2
Liver									
20:4ω6	31.1	28.6	25.4 ***	23.1	18.3 ***	13.2	12.6	7.9	9.2 ***
20:5ω3	0.1	0.1	0.5 ***	0.3	2.6 ***	2.9	6.9 ***	6.8	10.5 ***
22:5ω3	0.1	n.d.	n.d.	n.d.	0.8 ***	0.7	1.7 ***	1.1	2.0 ***
22:6ω3	3.7	9.3	11.9 ***	14.0	15.4 **	18.5	16.2 ***	21.1	16.5 ***
Σ ω3 PUFA	3.9	9.4	12.4	14.2	19.0 ***	22.1	25.1 **	29.0	29.6
Kidney									
20:4ω6	41.9	43.5	39.0	39.7	36.4	27.0	25.4	20.4	19.1
20:5ω3	0.01	0.2	0.5 **	0.8	3.4 ***	5.5	9.4 ***	17.2	16.7
22:5ω3	n.d.	n.d.	0.03 ***	n.d.	0.3 ***	0.2	0.5 ***	0.4	0.6 ***
22:6ω3	1.4	2.7	2.9	3.7	3.9	4.7	4.2	6.5	4.5 ***
Σ ω3 PUFA	1.4	2.9	3.4 *	4.5	7.8 ***	10.4	14.6 **	24.0	23.0
Muscle									
20:4ω6	22.6	16.3	16.4	11.1	11.8 ***	7.3	7.3	4.8	7.1
20:5ω3	0.03	0.06	0.2 **	0.2	0.7	0.6	1.4 ***	1.2	1.7 *
22:5ω3	0.6	0.5	1.0 ***	0.5	1.5 ***	0.7	1.3	0.8	1.5 **
22:6ω3	5.3	13.4	14.5	18.6	18.6	23.6	22.1	27.7	21.5 **
Σ ω3 PUFA	5.9	14.0	15.7	19.6	20.9 *	24.4	25.1	29.7	25.5 *
Testes									
22:5ω6	23.7	22.3	21.5	21.5	22.6	21.5	20.4 *	19.6	19.0
22:6ω3	0.5	1.1	1.4 **	1.7	1.9 **	2.6	2.67	3.5	3.1
Σ ω3 PUFA	0.5	1.1	1.4 **	1.7	2.1 **	2.7	3.0 **	3.8	3.5

^1^ At each supplementation level, significant differences between Control (DHA-Control) and Canola (DHA-Canola) are shown as * *p* < 0.05, ** *p* < 0.01, *** *p* < 0.0001 (student unpaired *t*-test). Mean values (n = 7–8). HOSO, high oleic sunflower seed oil.

## Data Availability

Data can be requested from the corresponding author due to technical limitations.

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
