# Peer review of "Sn1,3 Regiospecificity of DHA (22:6ω-3) of Plant Origin (DHA-Canola®) Facilitates Its Preferential Tissue Incorporation in Rats Compared to sn2 DHA in Algal Oil at Low Dietary Inclusion Levels"

_nutrients, 2025, doi:10.3390/nu17081306_

Round 1
Reviewer 1 Report
Comments and Suggestions for Authors
In this study, Belobrajdic and colleague compared the tissue incorporation between algal and plant DHA in rat. They demonstrated that transgenic plant-derived DHA is preferred to incorporate into various tissues. Although results are clear-cut, the authors must re-write the text by attention to the following points.
1. There are no ethical statements about animal experiments.
2. Table 1: There are many common parameters that are not necessary to include in this table.
3. Table 4 is difficult to discriminate. It is not necessary to show the results of all percentages, since the results are almost same. It is better to show summarized table.
4. Based on the above point, the authors must summarize the Results section of 3.5~10. It is very boring by many repetition.
5. Discussion section is too long and contains many unrelated parts to this study. The authors must totally re-write it.
Comments on the Quality of English LanguageSome correction is necessary.
Author Response
In this study, Belobrajdic and colleague compared the tissue incorporation between algal and plant DHA in rat. They demonstrated that transgenic plant-derived DHA is preferred to incorporate into various tissues. Although results are clear-cut, the authors must re-write the text by attention to the following points.
- There are no ethical statements about animal experiments.
Response: Ethical statement is report in the methods section of the manuscript, lines 97-100.
- Table 1: There are many common parameters that are not necessary to include in this table.
Response: To ensure full transparency and that each diet composition is balanced (ingredients add up to 1000 g/kg) we have included all ingredients in the Table as this is standard practice for Nutrition publications. Please advise if there is an alternative way that is preferred in this publication.
- Table 4 is difficult to discriminate. It is not necessary to show the results of all percentages, since the results are almost same. It is better to show summarized table.
Response: There are only 3 Tables in this manuscript. If you are referring to Table 3, we have only included the fatty acids most relevant to this manuscript (total of 6 fatty acids per tissue).
- Based on the above point, the authors must summarize the Results section of 3.5~10. It is very boring by many repetition.
Response: Results section 3.5-3.11 briefly summarises the FA data for the different tissues evaluated. This has been reported as succinctly as possible to ensure the main details of the findings are presented without it being too formulaic. We had considered discussing all of the tissue data together, however there are various differences between tissues which if discussed together would make it very confusing, especially given that some tissues have a very different FA composition and response to the other tissues (eg testes and brain).
- Discussion section is too long and contains many unrelated parts to this study. The authors must totally re-write it.
Response: The discussion has been reviewed and extensively edited to aid clearer discussion of the main outcomes of the study.
Reviewer 2 Report
Comments and Suggestions for Authors
Dear Authors;
Re: [Manuscript ID nutrients-3205269]
Manuscript title: "Sn1,3 regiospecificity of DHA (22:6ω-3) of plant origin (DHA-Canola®) facilitates its preferential tissue incorporation in rats compared to sn2 DHA in algal oil at low dietary inclusion levels"
The manuscript, describing an original in vivo study, aims to evaluate the bioavailability of a plant-based Docosahexaenoic acid oil in rats. You have employed a commercial DHA-Canola, and advocate that it performed better in comparison with the control samples. The article is well written and well presented in general.
Please find my questions / comments / suggestions below:
1. Abbreviations left undescribed in the Abstract and Introduction.
2. Control samples mentioned in the Abstract are somehow confusing (Lines 15-18). Please make it clear for readership exactly what controls you used in this study.
3. You claimed that: "The present study is the first animal trial to evaluate the ω-3 LCPUFA bioavailability of a plant-based DHA oil". However a search on Google Scholar using keywords such as: [ω-3 LCPUFA bioavailability of a plant-based DHA oil "animal studies"] retrieves 76 results and same search with ["in vivo"] retrieves 159 results.
4. Materials used in the study, their sources and information are missing.
5. Table 1 is not presented properly.
6. In Discussions section, you claimed that:
"The findings of this study clearly demonstrated that plant-based DHA-Canola is freely bioavailable and leads to similar outcomes as a commercially available marine-based oil high in DHA."
Is this claim supported by prior or present study data? If so, please clearly specify.
7. Another interesting aspect of your manuscript is the claim: "A highly novel and important finding from this study is that DHA-Canola fed animals had less fat mass compared with DHA-Control fed animals." While in the Conclusions you stated: "This 12-week feeding trial showed that rats consuming diets containing DHA-Canola had appreciable weight gain, ...".
8. Please clarify "appreciable" in the above sentence.
9. Any consideration / usefulness or comparison with nano-encapsulated DHA and EPA? https://doi.org/10.1016/j.foodchem.2012.07.016
and
https://doi.org/10.3109/08982104.2013.839702
10. Recommendations for future studies will benefit the readership. Please consider adding few lines in this context.
Thank you
Author Response
Dear Authors;
Re: [Manuscript ID nutrients-3205269]
Manuscript title: "Sn1,3 regiospecificity of DHA (22:6ω-3) of plant origin (DHA-Canola®) facilitates its preferential tissue incorporation in rats compared to sn2 DHA in algal oil at low dietary inclusion levels"
The manuscript, describing an original in vivo study, aims to evaluate the bioavailability of a plant-based Docosahexaenoic acid oil in rats. You have employed a commercial DHA-Canola, and advocate that it performed better in comparison with the control samples. The article is well written and well presented in general.
Please find my questions / comments / suggestions below:
- Abbreviations left undescribed in the Abstract and Introduction.
Response: It is our understanding that abbreviations for fatty acids such as DHA do not need to be defined given that we have provided the molecular structure after mentioning DHA and EPA for the first time in the Abstract and/or Introduction.
In addition, DHASCO® is a product name so it isn’t an abbreviation.
- Control samples mentioned in the Abstract are somehow confusing (Lines 15-18). Please make it clear for readership exactly what controls you used in this study.
Response: Abstract has been edited to clarify groups/treatments.
- You claimed that: "The present study is the first animal trial to evaluate the ω-3 LCPUFA bioavailability of a plant-based DHA oil". However a search on Google Scholar using keywords such as: [ω-3 LCPUFA bioavailability of a plant-based DHA oil "animal studies"] retrieves 76 results and same search with ["in vivo"] retrieves 159 results.
Response: Although there are many studies that arise from this search term, none of these publications involve a dietary intervention using “plant-based DHA oil.” All DHA oil commercially available is sourced from marine animals/organisms. This is what makes the current study highly novel.
- Materials used in the study, their sources and information are missing.
Response: We believe we have listed the sources of all materials used in this study. Eg dietary ingredients are listed in 2.2 (lines 130-136), the company and location for all scientific equipment is listed where first mentioned in the Methods section. If there are any specific details still missed please notify by line number.
- Table 1 is not presented properly.
Response: It is not clear what is wrong with the presentation of this Table. Please provide specific concerns with the Table presentation.
- In Discussions section, you claimed that:
"The findings of this study clearly demonstrated that plant-based DHA-Canola is freely bioavailable and leads to similar outcomes as a commercially available marine-based oil high in DHA."
Is this claim supported by prior or present study data? If so, please clearly specify.
Response: This first statement of the Discussion is supported by the subsequent sentences of that paragraph where we summarise the main outcomes of the study (line 360-372). We have removed the word “freely.”
- Another interesting aspect of your manuscript is the claim: "A highly novel and important finding from this study is that DHA-Canola fed animals had less fat mass compared with DHA-Control fed animals." While in the Conclusions you stated: "This 12-week feeding trial showed that rats consuming diets containing DHA-Canola had appreciable weight gain, ...".
Please clarify "appreciable" in the above sentence.
Response: The word “appreciable” has been removed as it was not appropriate.
- Any consideration / usefulness or comparison with nano-encapsulated DHA and EPA? https://doi.org/10.1016/j.foodchem.2012.07.016
and
https://doi.org/10.3109/08982104.2013.839702
Response: The nano-encapsulated lipids have potentially interesting applications. However, in food applications explored in the current study we show that the DHA-Canola has high bioavailability. An important next step is for the nano-encapsulated lipids is in vivo to evaluate bioavailability, tissue incorporation and functionality. This data could then be compared with the current study.
- Recommendations for future studies will benefit the readership. Please consider adding few lines in this context.
Response: At line 519 and 448 we suggest further studies are needed to explore the possible role of higher DPA intake on fat metabolism and reduced fat mass.
Thank you
Reviewer 3 Report
Comments and Suggestions for Authors
The present experimental study is not only interesting but also a testament to the researchers' hard work and dedication. The omega-3 long-chain polyunsaturated fatty acids (n-3 LC-PUFAs) have gained substantial interest due to their specific structure and biological functions. Humans cannot naturally produce these fatty acids (FAs), so obtaining them from the diet is crucial.
Moreover, an increasing body of evidence supports a link between low intakes of ω-3 long-chain polyunsaturated fatty acids (LCPUFA) and numerous diseases and health conditions.
The authors wrote,’’A highly novel and important finding from this study is that DHA-Canola fed animals had less fat mass than DHA-Control fed animals.’’. However, the authors should provide more clarity on the body weight change between the active and control groups, as it is not clear if there was a significant change.
The methodology is appropriate. The manuscript is well written, and the discussion/conclusions are acceptable.
Overall, data are of interest.
Comments on the Quality of English Languagenone
Author Response
The present experimental study is not only interesting but also a testament to the researchers' hard work and dedication. The omega-3 long-chain polyunsaturated fatty acids (n-3 LC-PUFAs) have gained substantial interest due to their specific structure and biological functions. Humans cannot naturally produce these fatty acids (FAs), so obtaining them from the diet is crucial.
Moreover, an increasing body of evidence supports a link between low intakes of ω-3 long-chain polyunsaturated fatty acids (LCPUFA) and numerous diseases and health conditions.
The authors wrote,’’A highly novel and important finding from this study is that DHA-Canola fed animals had less fat mass than DHA-Control fed animals.’’. However, the authors should provide more clarity on the body weight change between the active and control groups, as it is not clear if there was a significant change.
Response: All changes reported in the Discussion are significant. The mean values and significance values are all reported in the text of the Results section (lines 205-215).
The methodology is appropriate. The manuscript is well written, and the discussion/conclusions are acceptable.
Overall, data are of interest.
Reviewer 4 Report
Comments and Suggestions for Authors
The manuscript includes a practical study focused on the employment of DHA-canola for health enhancing in rats during an interventionary study.
I found it well presented, justified and discussed. However, some aspects could be performed.
Abstract
It is somewhat long, according to the current journal requirements.
Material and methods
Lines 147-165: Methods employed for biochemical determinations ought to be provided with more information, i.e., cholesterol, triacylglycerols, glucose, PL extraction, … The information included is really very short.
The word triglycerides ought to be replaced by triacylglycerols throughout the whole manuscript.
Line 151: The FAME quantification is not explained. Was there a quantitative standard employed ? I consider this is a very important concern as it is the basis of a great part of the study.
Discussion and Conclusions
Since the authors found that the two lowest DHA-canola levels (i.e., 0.3 and 1.0%) led to the most valuable results, the authors could discuss on the possibility of testing lower levels or even carry out an optimized study around this range.
Author Response
The manuscript includes a practical study focused on the employment of DHA-canola for health enhancing in rats during an interventionary study.
I found it well presented, justified and discussed. However, some aspects could be performed.
Abstract
It is somewhat long, according to the current journal requirements.
Response: The abstract is now 191 words.
Material and methods
Lines 147-165: Methods employed for biochemical determinations ought to be provided with more information, i.e., cholesterol, triacylglycerols, glucose, PL extraction, … The information included is really very short.
Response: The methods have been edited. As the methods are standardized by the company for routine clinical analysis of samples, the details we have provided are sufficient for others to replicate these analyses.
The word triglycerides ought to be replaced by triacylglycerols throughout the whole manuscript.
Response: Edited as suggested.
Line 151: The FAME quantification is not explained. Was there a quantitative standard employed ? I consider this is a very important concern as it is the basis of a great part of the study.
Response: The details of FAME analysis is report at line 153-167. Quantification details are included at line 163-165: “Peak identification was based on a comparison of retention times with Supelco 37-Component FAME Mix 47 885-U (Sigma-Aldrich). Individual fatty acids were calculated as a percentage of the total fatty acids.”
Discussion and Conclusions
Since the authors found that the two lowest DHA-canola levels (i.e., 0.3 and 1.0%) led to the most valuable results, the authors could discuss on the possibility of testing lower levels or even carry out an optimized study around this range.
Response: Following this study the most important next step is to trial equivalent doses in a clinical trial. Subsequently, this statement has been added to the conclusion (line 559-561).
Round 2
Reviewer 1 Report
Comments and Suggestions for Authors
Revision is insufficient and unfaithful. Basically, the manuscript was hardly improved. In particular, although the authors said 'The discussion has been reviewed and extensively edited`, there remains so much strange sentence. For example, what is the 'marine-based oil (Line 362)? In such case, marine algae oil is proper. `Dose-responsive manner (Line364)` is also improper, because this study never investigated the pharmacological responses. Sentence of line367-369 is too long. Based on these points, it was felt that the authors did not totally re-evaluate the manuscript.
Comments on the Quality of English LanguageThe quality of English language is too low. It is clear that the senior researchers never checked the manuscript.
Author Response
Reviewer 1: Comments and Suggestions for Authors
In this study, Belobrajdic and colleague compared the tissue incorporation between algal and plant DHA in rat. They demonstrated that transgenic plant-derived DHA is preferred to incorporate into various tissues. Although results are clear-cut, the authors must re-write the text by attention to the following points.
- There are no ethical statements about animal experiments.
Response: Ethical statement is report in the methods section of the manuscript, lines 97-100.
- Table 1: There are many common parameters that are not necessary to include in this table.
Response: To ensure full transparency and that each diet composition is balanced (ingredients add up to 1000 g/kg) we have included all ingredients in the Table as this is standard practice for Nutrition publications. Please advise if there is an alternative way that is preferred in this publication.
- Table 4 is difficult to discriminate. It is not necessary to show the results of all percentages, since the results are almost same. It is better to show summarized table.
Response: There are only 3 Tables in this manuscript. If you are referring to Table 3, we have only included the fatty acids most relevant to this manuscript (total of 6 fatty acids per tissue).
- Based on the above point, the authors must summarize the Results section of 3.5~10. It is very boring by many repetition.
Response: Results section 3.5-3.11 briefly summarises the FA data for the different tissues evaluated. This has been reported as succinctly as possible to ensure the main details of the findings are presented without it being too formulaic. We had considered discussing all of the tissue data together, however there are various differences between tissues which if discussed together would make it very confusing, especially given that some tissues have a very different FA composition and response to the other tissues (eg testes and brain).
- Discussion section is too long and contains many unrelated parts to this study. The authors must totally re-write it.
Response: The discussion has been reviewed and extensively edited to aid clearer discussion of the main outcomes of the study.
Reviewer 4 Report
Comments and Suggestions for Authors
The manuscript has been performed. However, some of the aspects previously mentioned to be revised are still as in the previous version.
Lines 147-149: The explanation of the analytical procedure of methods is not very scientific. Please, provide more information so that a possible reader can repeat the study.
Line 162: The quantification method for FAMEs is still not clarified. If a standard was not employed, which would have been desirable, the authors ought to express the way they did it. Areas provided by the GLC analysis ?
Author Response
Reviewer 4
The manuscript includes a practical study focused on the employment of DHA-canola for health enhancing in rats during an interventionary study.
I found it well presented, justified and discussed. However, some aspects could be performed.
Abstract
It is somewhat long, according to the current journal requirements.
Response: The abstract is now 191 words.
Material and methods
Lines 147-165: Methods employed for biochemical determinations ought to be provided with more information, i.e., cholesterol, triacylglycerols, glucose, PL extraction, … The information included is really very short.
Response: The methods have been edited. As the methods are standardized by the company for routine clinical analysis of samples, the details we have provided are sufficient for others to replicate these analyses.
The word triglycerides ought to be replaced by triacylglycerols throughout the whole manuscript.
Response: Edited as suggested.
Line 151: The FAME quantification is not explained. Was there a quantitative standard employed ? I consider this is a very important concern as it is the basis of a great part of the study.
Response: The details of FAME analysis is reported at lines 164-176. Quantification details are included at line 172-173: “Peak identification was based on a comparison of retention times with Supelco 37-Component FAME Mix 47 885-U (Sigma-Aldrich). Individual fatty acids were calculated as a percentage of the total fatty acids.”
Discussion and Conclusions
Since the authors found that the two lowest DHA-canola levels (i.e., 0.3 and 1.0%) led to the most valuable results, the authors could discuss on the possibility of testing lower levels or even carry out an optimized study around this range.
Response: Following this study the most important next step is to trial equivalent doses in a clinical trial. Subsequently, this statement has been added to the conclusion (line 569-571).
Round 3
Reviewer 1 Report
Comments and Suggestions for Authors
The authors never changed the manuscript.
Comments on the Quality of English LanguageThe quality is very low.
Author Response
I cant see any additional points that need to be addressed from this reviewer.